# *alms1* mutant zebrafish do not show hair cell phenotypes seen in other cilia mutants

**Lauren Parkinson, Tamara M. Stawicki** *

Neuroscience Program, Lafayette College, Easton, Pennsylvania, United States of America

* stawickt@lafayette.edu

**Data Availability Statement:** All relevant data are within the paper and its Supporting Information files.

**Funding:** The authors received no specific funding for this work.

## Abstract

Multiple cilia-associated genes have been shown to affect hair cells in zebrafish (*Danio rerio*), including the human deafness gene *dcdc2*, the radial spoke gene *rsph9*, and multiple intraflagellar transport (IFT) and transition zone genes. Recently a zebrafish *alms1* mutant was generated. The *ALMS1* gene is the gene mutated in the ciliopathy Alström Syndrome a disease that causes hearing loss among other symptoms. The hearing loss seen in Alström Syndrome may be due in part to hair cell defects as *Alms1* mutant mice show stereocilia polarity defects and a loss of hair cells. Hair cell loss is also seen in postmortem analysis of Alström patients. The zebrafish *alms1* mutant has metabolic defects similar to those seen in Alström syndrome and *Alms1* mutant mice. We wished to investigate if it also had hair cell defects. We, however, failed to find any hair cell related phenotypes in *alms1* mutant zebrafish. They had normal lateral line hair cell numbers as both larvae and adults and normal kinocilia formation. They also showed grossly normal swimming behavior, response to vibrational stimuli, and FM1-43 loading. Mutants also showed a normal degree of sensitivity to both short-term neomycin and long-term gentamicin treatment. These results indicate that cilia-associated genes differentially affect different hair cell types.

## Introduction

Hearing and balance disorders are common sensory disorders [1, 2]. There is a significant genetic component to these disorders, with over 50% of congenital hearing loss seen in newborns being hereditary [3]. To date, 121 nonsyndromic hearing loss genes have been identified in addition to many syndromic hearing loss genes [4]. While most of these genes are only implicated in hearing loss, mutations in a small subset also cause balance disorders [5, 6]. Genetic mutations causing hearing loss can affect various processes important for hearing. A number of these mutations specifically affect the development and function of the sensory hair cells responsible for hearing [3].

Zebrafish have evolved as a useful model to study the genetics of hair cell function. Zebrafish are powerful genetic models and have hair cells on the surface of their bodies as part of the lateral line system, which allows for the observation of hair cells in an intact *in vivo* model. Researchers have carried out large-scale genetic screens to identify zebrafish mutants with auditory or vestibular defects [7–9] as well as generated zebrafish models of known deafness genes [10, 11].

**Competing interests:** The authors have declared that no competing interests exist.

One class of genes that have been shown to influence hair cells in humans and zebrafish are cilia-associated genes. Multiple ciliopathies, genetic disorders with mutations in cilia-associated genes, have been associated with hearing loss, including Usher Syndrome, Alström Syndrome, and Bardet-Biedl Syndrome (BBS) [12–16]. A number of Usher Syndrome genes have also been shown to affect hair cell function in zebrafish [17–22]. Other cilia genes associated with hearing loss in humans include *DCDC2*, *CDC14A*, *CCDC114*, and the basal body genes *CEP78* and *CEP250* [23–28]. While some of these genes, such as *dcdc2*, are similarly necessary for hair cell function in zebrafish [26] others such as *cdc14A* do not cause zebrafish hair cell phenotypes when mutated [29]. There have also been a number of cilia-associated genes that have not been implicated in human deafness but have been shown to cause hair cell phenotypes when disrupted in zebrafish. Mutations in intraflagellar transport (IFT) genes, genes necessary for the transport of proteins along cilia, in zebrafish have been shown to decrease hair cell numbers, lead to resistance to aminoglycoside-induced hair cell death, and to cause defects in FM1-43 uptake and early reverse polarity hair cell mechanotransduction activity [30–33]. Mutations in transition zone genes, genes that act as gatekeepers for proteins exiting and entering cilia, have also been shown to cause resistance to aminoglycoside-induced hair cell death but to have normal FM1-43 uptake and control hair cell numbers [32, 34], and a mutation in *rsph9*, a radial spoke gene in motile cilia, caused a reduced initiation of the startle response to acoustic stimuli in zebrafish suggesting a potential hair cell defect [35].

Recently a zebrafish *alms1* mutant line was generated [36]. *ALMS1* is the gene responsible for Alström Syndrome, a ciliopathy characterized by hearing and vision loss in addition to obesity and diabetes [13, 37, 38]. Dilated cardiomyopathy, hypertriglyceridemia, gastrointestinal disturbances, neurological symptoms, and liver and kidney dysfunction have also been observed in some patients [39]. ALMS1 localizes to the basal body at the base of cilia leading to the classification of Alström Syndrome as a ciliopathy [40]. Hearing loss in Alström Syndrome patients usually begins in childhood and is progressive [39, 41, 42] with patients also showing abnormal distortion product otoacoustic emissions (DPOAEs) [42, 43] Vestibular defects have not been observed. Postmortem analysis of patient auditory tissue shows degeneration of the organ of Corti, including a loss of hair cells. Additionally, there is atrophy of the stria vascularis and spiral ligament and degeneration of the spiral ganglion. In contrast to the auditory tissue, vestibular epithelium looked normal in these patients, in agreement with the lack of vestibular symptoms [44]. Mouse *Alms1* mutants show similar phenotypes as human patients, including obesity, retinal dysfunction, hyperinsulinemia, and delayed onset hearing loss [45–47]. Hair cells in *Alms1* mutant mice show misshapen stereocilia bundles and polarity defects and a loss of outer hair cells with age. Strial atrophy is also seen similar to what was observed in human postmortem samples [44, 47]. Similar to mouse mutants, zebrafish *alms1* mutants share many phenotypes with Alström Syndrome patients, including retinal degeneration, kidney and cardiac defects, increased fat disposition in the liver, a propensity for obesity, hyperinsulinemia, and glucose response defects [36]. However, hair cells and the auditory and vestibular systems of these fish have not been examined.

We wished to examine *alms1* mutant zebrafish to see if they showed hair cell phenotypes similar to mammalian *Alms1* mutants and other zebrafish cilia-associated gene mutants. We found that *alms1* mutant zebrafish had normal cilia formation in lateral line hair cells and other ciliated cells similar to what has been observed in mammalian *Alms1* mutants. We also failed to find any hair cell phenotypes in these mutants. They showed grossly normal audiovestibular behavior and sensitivity to aminoglycosides as both larvae and adults. They also showed normal FM1-43 uptake into hair cells. These results and the lack of vestibular defects seen in mammals following *Alms1* mutations suggest a specific role for *Alms1* in the auditory system rather than more global hair cell function. It also shows that cilia-associated genes have distinct roles in different hair cell types.

## Materials and methods

### Animals

Experiments used either 1 or 5 days post-fertilization (dpf) *Danio rerio* (zebrafish) larvae or adult zebrafish over three months of age. We used the previously described *alms1*$^{umd2}$ mutant line, which causes a premature stop codon [36]. Mutant larvae were generated by either crossing two heterozygous animals together, or one heterozygous animal to a homozygous mutant, and comparing homozygous mutants to either wild-type or heterozygous siblings born at the same time. For the experiments on vibrational stimulus response and adult zebrafish, homozygous mutants were incrossed to generate the mutant animals, and homozygous wild-type siblings of those mutants were crossed to heterozygous wild-type siblings to generate the wild-type controls. Larvae were raised in embryo media (EM) consisting of 1 mM MgSO$_4$, 150 μM KH$_2$PO$_4$, 42 μM Na$_2$HPO$_4$, 1 mM CaCl$_2$, 500 μM KCl, 15 mM NaCl, and 714 μM NaHCO$_3$ and housed in an incubator maintained at 28.5°C with a 14/10 hour light/dark cycle. The Lafayette College Institution Animal Care and Use Committee approved all experiments.

### Antibody labeling, imaging, and cilia length measurement

Zebrafish larvae and fins used for antibody labeling were fixed for two hours at room temperature in 4% paraformaldehyde. Antibody labeling was carried out as previously described [48]. Cilia were labeled with a mouse anti-acetylated tubulin primary antibody (Millipore-Sigma, T7451) diluted at 1:1,000 in antibody block. Larvae used for hair cell counts were labeled with a rabbit anti-parvalbumin primary antibody (ThermoFisher, PA1-933) diluted at 1:1,000 in antibody block. Fins used for hair cell counts were labeled with both the parvalbumin antibody and a mouse anti-otoferlin antibody (Developmental Studies Hybridoma Bank, HCS-1) diluted at 1:100 in antibody block.

Images of antibody labelled larvae and fins were taken on a Zeiss LSM800 confocal microscope. A stack of images 1 μm apart were taken throughout the tissue of interest and then maximum projection images of these stacks were generated for inclusion in the figures. To measure cilia length we chose a representative cilia that we could easily trace in each image and measured the length using Fiji.

### Vibrational stimulus response

To test if *alms1* mutants would respond to vibrational stimuli we tapped a petri dish containing 50 *alms1* mutants with a glass pipette and recorded their response with an iPhone SE. To quantify this behavior we used a modified version of methods previously described [49]. Fish were moved into a fresh petri dish 8 fish at a time. After giving the fish a few minutes to acclimate to the new dish the dish was taped with a glass pipette three times a few seconds apart while recording the response of the fish with an iPhone SE. Videos were then scored with each fish given a score of 0 for no movement in response to the tap and a score of 1 if they moved in response to the tap. Fish that were showing spontaneous movement at the times of any of the taps or that were on the edge of the dish during the experiment were excluded. An average score across the three experiments was generated for each fish.

### FM1-43 uptake

Fish were treated with 2.25 μM FM 1-43FX (ThermoFisher, F35355) for 1 minute in EM, washed 3 times in EM, and then anesthetized with MS-222/Tricaine-S (Pentair Aquatic Eco-Systems, TRS1) for imaging. Fish were imaged on a Zeiss LSM800 confocal microscope. For each fish, a single neuromast was imaged by taking a stack of 5 1 μm optical sections. Image

analysis was carried out in Fiji. A maximum projection image was made of each stack of images, and the average fluorescent intensity of the cell bodies of the entire neuromast was measured. Additionally, the average fluorescent intensity of an area in the background of the image was measured. Finally, the value for the fluorescence of the neuromast was divided by the value of the background fluorescence. Larvae were euthanized and genotyped after imaging.

### Aminoglycoside treatment and hair cell counts

Fish were treated with either neomycin solution (Sigma-Aldrich, N1142) or gentamicin solution (Sigma-Aldrich G1272) at the indicated doses diluted in EM. For all neomycin treatment experiments, fish were treated with neomycin for 30 minutes, washed 3 times in either plain EM for larvae or plain system water for adult fish, left to recover in the third wash for one hour, and then tissue was collected for antibody labeling. In the case of larvae, whole larvae were fixed, whereas for adult fish, the fish were anesthetized, and their caudal fins were amputated and fixed. For gentamicin treatment, larvae were treated with gentamicin for 6 hours, washed 3 times in plain EM, and then fixed for antibody labeling. For adult zebrafish, 6 neuromasts on each fin were counted using the HCS-1 stain, and an average hair cell/neuromast number was generated for each animal. For larvae, the OP1, M2, IO4, MI2, and MI1 neuromasts [50] were counted using the parvalbumin stain, and again an average number of hair cells/neuromast number was generated for each animal. Larvae were genotyped after counting was complete. All hair cell counts were carried out on an Accu-Scope EXC-350 microscope.

### Statistical analysis

All statistics were calculated using GraphPad Prism software (version 6.0).

## Results

### Cilia morphology is normal in *alms1* mutants

There have previously been conflicting results regarding whether ALMS1 plays a role in cilia formation and maintenance. Cilia in *Alms1* mutant mice and fibroblasts isolated from Alström syndrome patients appear grossly normal [40, 46, 51–53] even when there is no visible antibody labeling for ALMS1 protein [52]. However, RNAi knockdown of *Alms1* in cultured cells results in abnormal and stunted cilia [53, 54]. To test what cilia morphology looked like in zebrafish *alms1* mutants, we stained 24 hours post fertilization (hpf) and 5dpf zebrafish larvae with acetylated tubulin. We found that cilia morphology looked grossly normal in both early hair cells of the otic vesicle, hair cells of the lateral line and cells of the olfactory pit in *alms1* mutants (Fig 1) We also failed to see any early hair cells of the otic vesicle or neuromasts that were missing cilia or obvious gaps of missing cilia in the olfactory pit. Measuring cilia length we did not see any significant differences in the average cilia length in these tissues. We did however, observe an increase in the variability of cilia length in *alms1* mutants in the early hair cells in the otic vesicle (Fig 1).

### *alms1* mutants do not show mechanotransduction defect phenotypes

Zebrafish mechanotransduction mutants show several characteristic behavioral phenotypes, including a failure to remain upright, circling behavior when swimming, and a failure to respond to acoustic-vibrational stimuli [9]. It has previously been shown that zebrafish morphants of another cilia-associated deafness gene, *dcdc2*, show similar behavioral defects [26]. We, however, failed to observe any of these phenotypes in *alms1* mutant zebrafish. *alms1* 5dpf

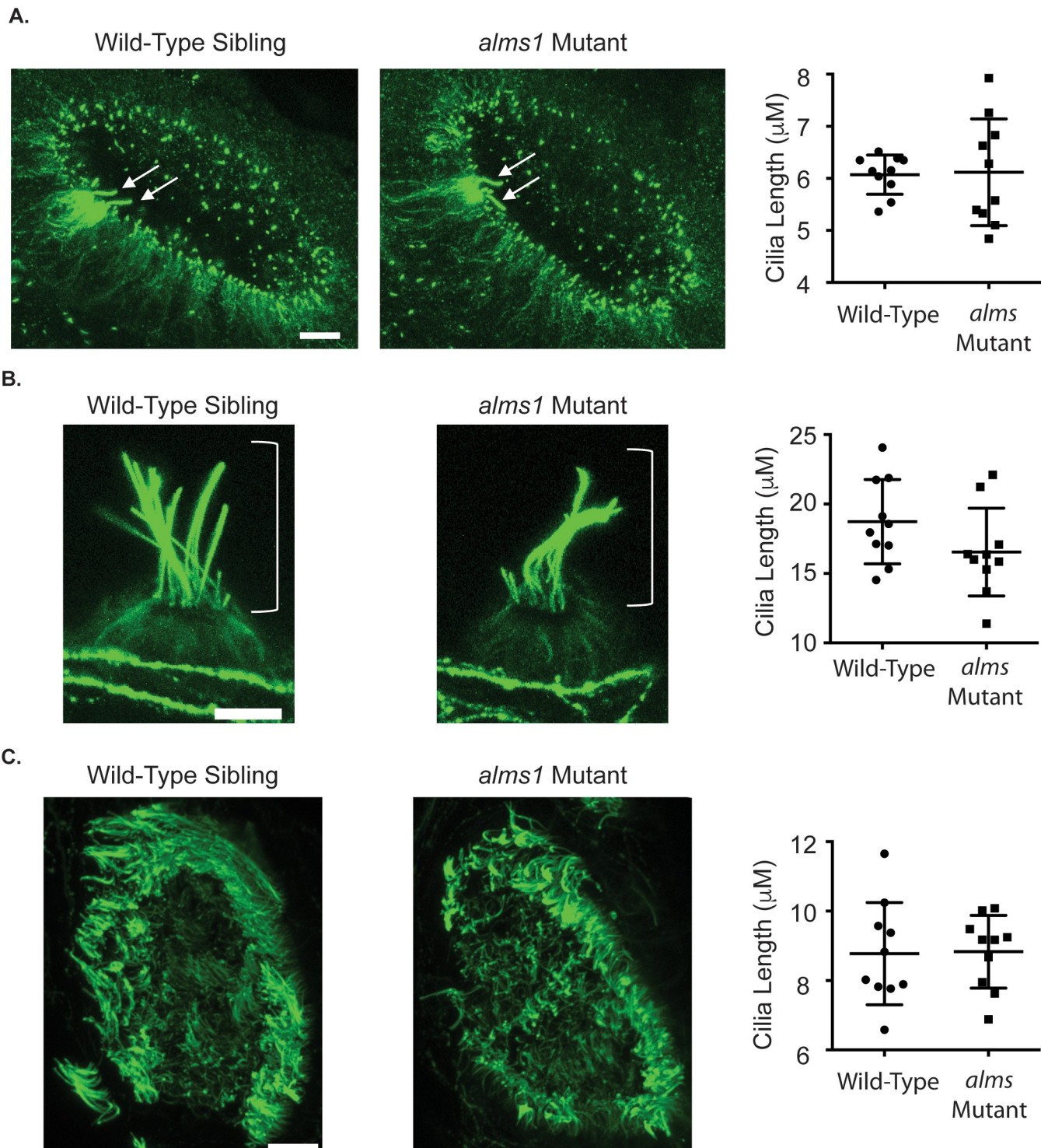

**Fig 1. Cilia formation is largely normal in *alms1* mutants.** (A) Representative images of the otic vesicle of a 24 hpf wild-type (left) and *alms1* mutant (center) zebrafish embryo stained with acetylated tubulin. The arrows point to the cilia of the early hair cells. To the right is quantification of cilia length. There was no significant difference in average kinocilia length between wild-type and mutant zebrafish, however, there was more variation in length seen in *alms1* mutants. (p = 0.0065 by F test to compare variances) (B) Representative images of the IO1 neuromast stained with acetylated tubulin in wild-type (left) and *alms1* mutant (center) 5dpf zebrafish larvae. The brackets show the position of the kinocilia. Kinocilia appear grossly normal in *alms1* mutant fish. To the right is quantification of IO1 kinocilia length. There was no significant difference between wild-type and mutant zebrafish (p = 0.1312 by an unpaired t-test) (C) Representative images of the olfactory pit stained with acetylated tubulin in wild-type (left) and *alms1* mutant (center) 5dpf zebrafish larvae. Again cilia appear

grossly normal in *alms1* mutants. To the right is quantification of cilia length in the olfactory pit. There was no significant difference between wild-type and mutant zebrafish (p = 0.9247 by unpaired t-test). Scale bar for all images = 10 μm. Quantification graphs show individual data points along with the mean and standard deviation of the data, n = 10 fish per group.

mutant larvae remain upright, and those without body morphology defects showed a grossly normal escape response following a vibrational stimulus generated by taping on their dish (S1 Movie and Fig 2). Adult homozygous mutants also showed normal swimming behavior (S2 Movie). The rapid uptake of FM1-43 is also reduced or eliminated when hair cell mechano-transduction activity is impaired [55–57]. It has previously been shown that other cilia gene mutants, specifically IFT mutants, show reduced levels of FM1-43 loading into lateral line hair cells [32, 33]. However, we failed to observe any significant differences in FM1-43 loading in *alms1* mutants compared to wild-type siblings (Fig 3).

### *alms1* mutants are not resistant to aminoglycoside-induced hair cell death

It has previously been shown that some zebrafish cilia-associated gene mutants are resistant to aminoglycoside-induced hair cell death, specifically transition zone, including the basal body gene *cep290*, and IFT gene mutants [32–34]. To test if this was also the case for *alms1* mutants, we used both short-term neomycin and long-term gentamicin treatment paradigms and

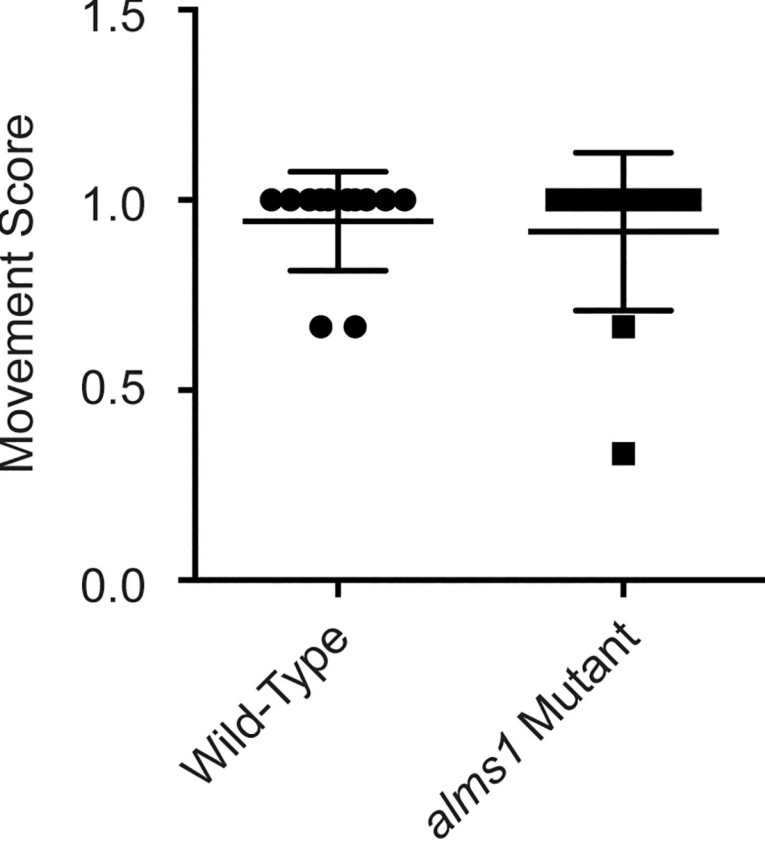

**Fig 2. *alms1* mutants respond to vibrational stimuli.** Quantification of the number of times fish respond to a vibrational stimuli across three times. A score of 1 means a fish responded all three times whereas a score of 0 means a fish never responded. There was no significant difference between the two groups by an unpaired t-test (p = 0.6977), n = 12 fish per group. Individual data points are shown along with the mean and standard deviation of the data.

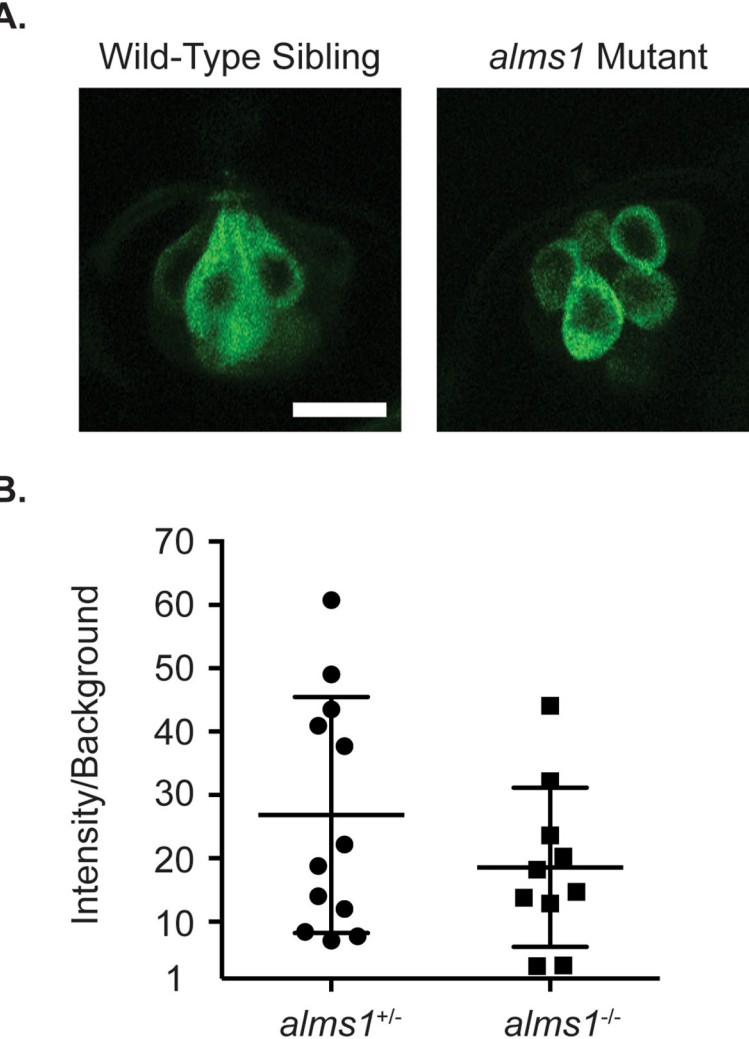

**Fig 3. FM1-43 loading is normal in *alms1* mutants.** (A) Representative images of neuromasts from wild-type siblings (left) and *alms1* mutants (right) treated with FM1-43. Scale bar = 10 μm. (B) Quantification of the fluorescent intensity of FM1-43 in neuromasts of heterozygous wild-type siblings and homozygous *alms1* mutants. There was no significant difference between the two groups by an unpaired t-test (p = 0.2474). Individual data points are shown along with the mean and standard deviation of the data.

quantified hair cell number in 5dpf zebrafish larvae. We failed to see any significant differences between wild-type mutants and homozygous mutants in either control conditions or following aminoglycoside treatment (Fig 4A–4C). Many cilia-associated genes are maternally expressed in zebrafish, meaning a heterozygous mother loads wild-type RNA into the egg, and this can mask mutant phenotypes early in development [58–60]. Therefore, we wished to test if this was happening in *alms1* mutants by generating mutant larvae out of both heterozygous and homozygous mutant mothers and treating them with neomycin. We again failed to find any significant differences between the different genotypes (Fig 4B).

In human Alström syndrome, patients' hearing loss is usually not seen until later in childhood [39, 42, 61]. Likewise, hearing defects in *Alms1* mutant mice show a delayed onset [45–47]. Therefore, we wished to see if adult zebrafish were resistant to aminoglycoside-induced hair cell death even though larvae were not. We treated fish slightly over 3 months of age that

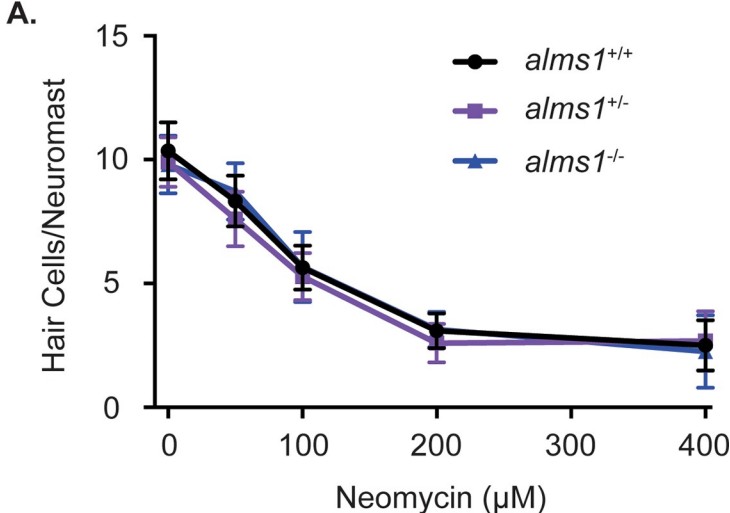

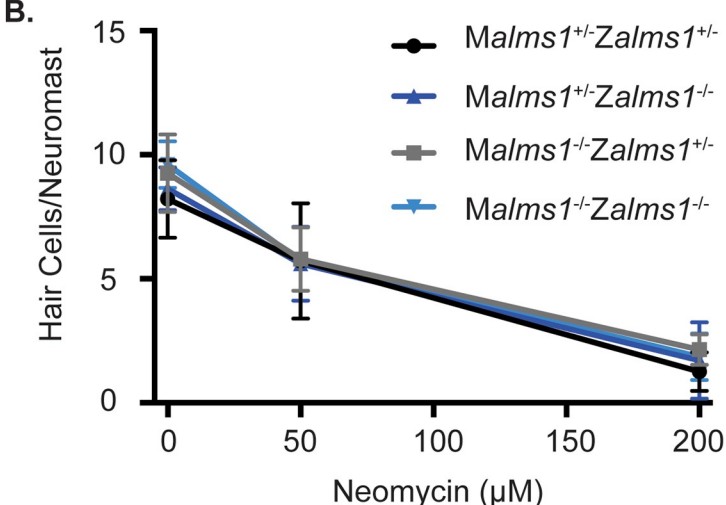

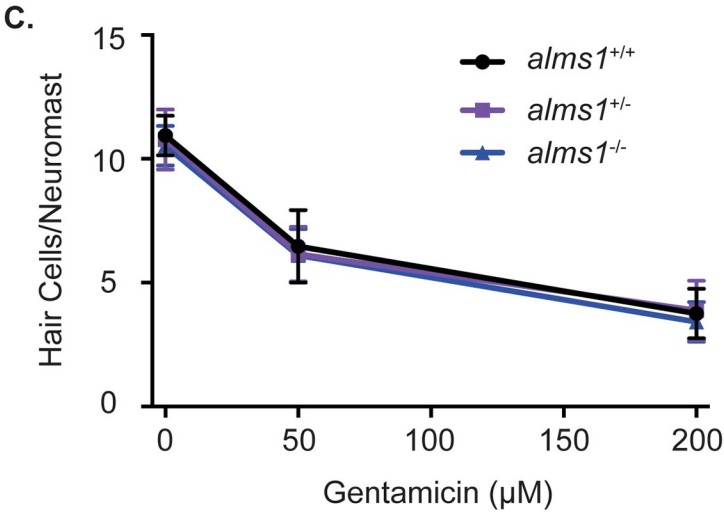

**Fig 4. *alms1* mutant larvae are not resistant to aminoglycoside-induced hair cell death.** (A) Dose-response curve for wild-type siblings and *alms1* mutants from an incross of heterozygous parents treated with 0–400 μM neomycin for 30 minutes with a one-hour recovery time. n = 7–19 fish per neomycin dose for homozygous wild-type and mutants and 20–25 for heterozygous fish. There were no significant differences due to genotype (p = 0.1273) or the interaction between genotype and neomycin (p = 0.3939) by two-way ANOVA. (B) Dose-response curve for fish that were either heterozygous or homozygous for the *alms1* mutation from mothers also either heterozygous or homozygous for the *alms1* mutation treated with 0–200 μM neomycin for 30 minutes with a one-hour recovery time. M = maternal genotype and Z = zygotic genotype. n = 7–16 fish per neomycin dose for each genotype. There were no significant differences due to genotype (p = 0.8060) or the interaction between genotype and neomycin (p = 0.2024) by two-way ANOVA. (C) Dose-response curve for wild-type siblings and *alms1* mutants from parents heterozygous for the *alms1* mutation treated with 0–200 μM gentamicin for 6 hours. n = 10–14 fish per gentamicin dose for homozygous wild-type and mutants and 21–25 for heterozygous fish. There were no significant differences due to genotype (p = 0.9401) or the interaction between genotype and neomycin (p = 0.2850) by two-way ANOVA. Data are shown as mean +/- standard deviation.

appeared to be mature adults (S2 Movie) with neomycin. We again failed to find any significant difference between homozygous mutants and related wild-type fish in either control lateral line hair cell numbers or hair cell numbers following neomycin treatment (Fig 5).

## Discussion

Zebrafish have emerged as a useful model to study hair cell function and death in part due to the external location of hair cells in the lateral line system. Multiple genes associated with both syndromic and nonsyndromic forms of deafness have been shown to cause hair cell related phenotypes in zebrafish [10, 11]. This is particularly true for genes affecting mechanotransduction or synaptic activity [62, 63]. However, not all deafness genes have been shown to have phenotypes when mutated in zebrafish, including the cilia-associated gene *cdc114* [29]. We found this also to be the case with *alsm1*. *alms1* has previously been shown to be expressed in hair cells of zebrafish larvae and adults [64, 65] as well as showing up in the mature lateral line hair cell cluster in single cell RNA-Seq data [66]. Despite this we noticed no obvious hair cell phenotypes in *alms1* zebrafish mutants. Fish showed grossly normal swimming behavior and response to vibrational stimuli, as well as normal hair cell number in contrast to what has previously been shown in mouse *Alms1* mutants [47] and human patients [44]. These mutants also showed normal FM1-43 uptake and aminoglycoside toxicity of lateral line hair cells in contrast to what has been seen in other zebrafish cilia-associated gene mutants [32–34].

There are multiple explanations for the lack of phenotypes we observed. First, it has previously been shown that CRISPR mutants can show genetic compensation that is not present when genes are merely knocked down [67]. It is also possible that the lack of phenotype we observe is due to a lack of sequence conservation between the mammalian and zebrafish *alms1* gene. The majority of the N-terminus of the human *ALMS1* gene, encoding for the tandem repeat domain of the protein, is specific to mammals [68, 69] with the first 1560 amino acids of the human gene having no homologous region in the zebrafish gene. However, as the zebrafish mutant shows multiple expected phenotypes due to defects in other tissues [36] if any of these issues were responsible for the lack of phenotype we see, it would have to be relatively hair cell specific. It is also possible that *alms1* has an ohnologue compensating for its loss in hair cells. Teleost fish underwent a genome duplication event [70, 71] and other zebrafish hair cell gene mutants have failed to completely phenocopy mammalian mutants due to either ohnologue specific functions or restricted expression of ohnologues [19, 72–74]. *alms1*, however, does not have any known ohnologues in zebrafish [75] and performing a protein blast for the human and zebrafish *alms1* gene as well as the ALMS protein motif did not identify any candidate ohnologues. This blast did show the presence of a *C10orf90* homologue in zebrafish, which is a known centrosomal gene with an ALMS motif [37, 68], however, this gene is also present in

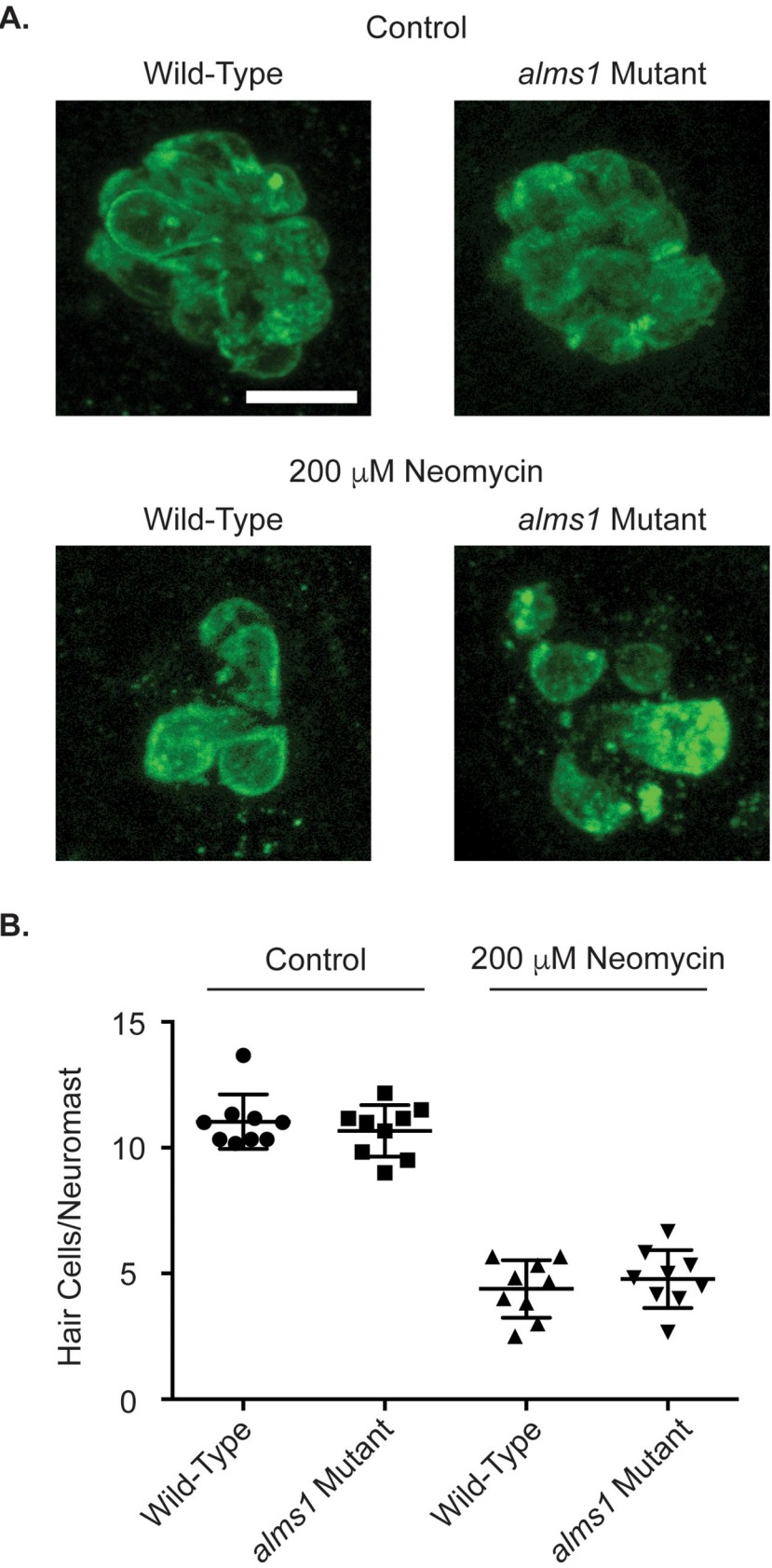

**Fig 5. Adult *alms1* mutant fish are not resistant to neomycin-induced hair cell death.** (A) Representative images of neuromasts from control (top) and neomycin treated (bottom) fish from wild-type (left) and *alms1* mutant (right) fish. (B) Quantification of hair cells/neuromast either in control conditions or following treatment with 200 μM neomycin. Individual data points are shown along with the mean and standard deviation. There were no significant differences due to genotype (p = 0.3073) or the interaction between genotype and neomycin (p = 0.9784) by two-way ANOVA.

mammals and is not able to compensate for *alms1* defects there. We can also not rule out more subtle defects, particularly in inner ear hair cells, that were not observed by the methods we used.

However, we feel the most likely explanation for the lack of observed phenotypes is different roles for *alms1* in different hair cell types. While human Alström patients and *Alms1* mutant mice show hearing defects and defects in auditory hair cells, there have not been any reported vestibular defects. ALMS1 specifically localizes to the centrosome or basal body of cilia [40, 47] and other basal body proteins have likewise been associated with hearing loss but not vestibular dysfunction [23, 24]. While different hair cell types have many shared genes, there are also many genes that are auditory or vestibular hair cell specific [76, 77]. Comparing lists of highly expressed zebrafish hair cell genes [64, 65] to these lists show overlap with several vestibular hair cell specific genes but no auditory hair cell specific genes. This suggests zebrafish hair cells are more closely related to mammalian vestibular hair cells than auditory hair cells. One of the defects seen in *Alms1* mutant hair cells in mammals, is stereocilia polarity [47]. This defect has also been shown in auditory hair cells of other cilia-associated gene mutants [15, 78–80]. However, in some cases, these mutants had no polarity defects in vestibular hair cells [79, 80] and these polarity defects have not been seen in lateral line hair cells of zebrafish cilia-associated gene mutants [30, 32]. Therefore, it appears that in mammalian auditory hair cells where the kinocilia, the true cilia of the cell, is lost early in development [81] the primary role of cilia-associated genes has evolved to be determining stereocilia polarity early in development. Whereas, in mammalian vestibular hair cells and zebrafish hair cells, where the kinocilia are maintained, these genes may play other functions that are not yet fully elucidated.

## Supporting information

**S1 Movie. alms1 mutant larvae's response to vibrational stimuli.**
(MP4)

**S2 Movie. Adult alms1 mutants.**
(MP4)

**S1 Data.**
(ZIP)

## Acknowledgments

We thank Jessica Nesmith and Norann Zaghloul for the *alms1* mutant strain, Ivan Cruz for advice on labeling hair cells in adult zebrafish, and Amy Badillo for assistance with zebrafish care.

## Author Contributions

**Formal analysis:** Lauren Parkinson, Tamara M. Stawicki.

**Investigation:** Lauren Parkinson, Tamara M. Stawicki.

**Methodology:** Lauren Parkinson, Tamara M. Stawicki.

**Project administration:** Tamara M. Stawicki.

**Supervision:** Tamara M. Stawicki.

**Visualization:** Tamara M. Stawicki.

**Writing – original draft:** Tamara M. Stawicki.

**Writing – review & editing:** Tamara M. Stawicki.

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
