## [Decision Letter · Decision Letter 0]

13 Jan 2021

PONE-D-20-38618

alms1 mutant zebrafish do not show hair cell phenotypes seen in other cilia mutants

PLOS ONE

Dear Dr. Stawicki,

Thank you for submitting your manuscript to PLOS ONE. After careful consideration, we feel that it has merit but does not fully meet PLOS ONE’s publication criteria as it currently stands. Therefore, we invite you to submit a revised version of the manuscript that addresses the points raised during the review process.

We look forward to receiving your revised manuscript.

Kind regards,

Hemant Khanna

Academic Editor

PLOS ONE

Journal Requirements:

2. We note that you are reporting an analysis of a microarray, next-generation sequencing, or deep sequencing data set. PLOS requires that authors comply with field-specific standards for preparation, recording, and deposition of data in repositories appropriate to their field. Please upload these data to a stable, public repository (such as ArrayExpress, Gene Expression Omnibus (GEO), DNA Data Bank of Japan (DDBJ), NCBI GenBank, NCBI Sequence Read Archive, or EMBL Nucleotide Sequence Database (ENA)). In your revised cover letter, please provide the relevant accession numbers that may be used to access these data. For a full list of recommended repositories, see http://journals.plos.org/plosone/s/data-availability#loc-omics or http://journals.plos.org/plosone/s/data-availability#loc-sequencing.

Reviewers' comments:

Reviewer's Responses to Questions

**Comments to the Author**

1. Is the manuscript technically sound, and do the data support the conclusions?

Reviewer #1: Partly

Reviewer #2: Yes

2. Has the statistical analysis been performed appropriately and rigorously? 

Reviewer #1: Yes

Reviewer #2: Yes

3. Have the authors made all data underlying the findings in their manuscript fully available?

Reviewer #1: Yes

Reviewer #2: Yes

4. Is the manuscript presented in an intelligible fashion and written in standard English?

Reviewer #1: Yes

Reviewer #2: Yes

5. Review Comments to the Author

Reviewer #1: Parkinson and Stawicki present data that suggest loss of alms1 in zebrafish does not affect vestibular hair cell function. Mutations in Alms1 are associated with Alstrom Syndrome in humans, which is a ciliopathy associated with loss of hearing and vision, as well as heart disease and diabetes. The zebrafish mutant was previously shown to have several phenotypes consistent with Alstrom Syndrome but the effects on hair cells had not been characterized. Here, the authors use imaging and functional assays to show that hair cell development and function in the lateral line were grossly normal.

Although brief, the manuscript is well-written and the figures are clearly presented and organized. The results are somewhat unexpected, given that prior studies has demonstrated functional conservation of alms1 in other zebrafish tissues. This study adds to the body of knowledge for alms1 function, but a few weaknesses remain. The zebrafish lateral line is considered an easily accessible tissue for vestibular hair cell study, but it may not fully recapitulate phenotypes limited to auditory hair cells. As such, the lack of a phenotype in the lateral line must be interpreted conservatively. The authors would be well served to consider additional experiments on auditory hair cells to fully rule out any possible phenotypes.

Major concerns

• The authors demonstrate that zebrafish alms1 mutants lack phenotypes typically associated with hair cell dysfunction and speculate that genetic compensation, lack of conservation with the mammalian homolog, or for a different role in different hair cell types. What the authors have not done, however, is show that alms1 is expressed in lateral line hair cells during development. If alms1 isn’t expressed in lateral line hair cells, then the absence of a phenotype would not be surprising.

• More information should also be included that discusses potential paralogs, including a rigorous homology search for any paralog/homologs that may not be listed in Ensembl.

• Lines 170-172: While the authors state that cilia look grossly normal, additional quantitative metrics would strengthen their argument. Are cilia lengths normal? Is there some way to assess cilia polarity or the degree of splaying that occurs in other mutants? While the authors are likely correct, having additional evidence that is less subjective would add rigor.

• To more fully support the argument that alms1 may have a role in auditory hair cells, the authors should examine cilia during auditory hair cell development or maintenance in alms1 larvae by imaging cilia within the otic vesicle (see early work from Jarema Malicki or Bruce Riley for examples).

Minor concerns

• Line 127: please correct the dilution value for the otoferlin antibody.

Reviewer #2: Full Title: alms1 mutant zebrafish do not show hair cell phenotypes seen in other cilia mutants

Since Alms1 is a basal body protein which mutations have been involved in Alström Syndrome patient’s hearing loss, the aim of this study was to elucidate the effect of Alsm1 mutation on hair cell phenotypes and function in the previously described Alsm1 mutant zebrafish.

The paper focused on hair cells morphology via staining in Alms1 zebrafish larvae and didn’t observed any morphology change. To study the effect of such mutation on hair cell properties, authors conducted escaping reflex experiments and vital dye staining (FM1-43) and showed no impact on mechanotransduction function in Alms1 mutant hair cells. Finally, authors conducted hair cell death analysis after neomycin or gentamicin treatment in adult and zebrafish larvae and couldn’t observe any increase in death sensitivity due to the loss of Alms1. This study concluded that Alms1 mutant zebrafish does not show hair cell defects.

This data suggests that Alms1, one of the major cilia-associated protein involve in human auditory hear cells does not show any phenotype in zebrafish hair cells, convincing that the Alsm1 mutant zebrafish might not be a proper model to study Alms1 auditory function. I consider the manuscript valuable and acceptable for publication with minor revisions:

- Even if the final message of this publication is the lack of phenotype, one concern is the absence of additional analysis. As an example, in figure 1, I would appreciate to know the number of ciliated cells even if change will not be observed. Also, authors should draw arrows to point the lateral line and cells of the olfactory pit in both control and Alms1 mutant zebrafish.

- Microscopy material and method details associated with the antibody staining are lacking in the method paragraph. Number of cells/cilia counted should also be precise.

- Escaping approach used to study mechanostransduction is not described in the method and no quantification of the escaping movement is provided with the supplemental video.

- Discussion paragraph on the lack of conservation with the N terminal mammalian isoform is pretty unclear and might need to be clarified or reformulated.

- Final paragraph questioning the potential specific auditory vs vestibular cell type expression and function bring an interesting point on the role of Alms1 and the importance to use the proper animal model to study auditory functions.

6. PLOS authors have the option to publish the peer review history of their article (what does this mean?). If published, this will include your full peer review and any attached files.

Reviewer #1: No

Reviewer #2: No

---

## [Author Response · Author response to Decision Letter 0]

25 Jan 2021

Below please find our response to the concerns brought up by the reviewers. We have put our responses in blue.

Reviewer #1

Major concerns

• The authors demonstrate that zebrafish alms1 mutants lack phenotypes typically associated with hair cell dysfunction and speculate that genetic compensation, lack of conservation with the mammalian homolog, or for a different role in different hair cell types. What the authors have not done, however, is show that alms1 is expressed in lateral line hair cells during development. If alms1 isn’t expressed in lateral line hair cells, then the absence of a phenotype would not be surprising.

We have investigated existing transcriptome data from zebrafish hair cells to confirm alms1 expression. We found it was identified as being expressed in studies that looked at the total hair cell transcriptome in both larvae and adults and also clustered with mature hair cells in a lateral line specific single cell RNA dataset. We have added this information to the discussion (lines 313-316).

• More information should also be included that discusses potential paralogs, including a rigorous homology search for any paralog/homologs that may not be listed in Ensembl.

Searching for known ohnologues of alms1 that may have resulted from the genome duplication in teleost fish did not identify any hits. We also performed searches for homologous proteins and found only one, a C10orf90 homologue which is also a known mammalian ALMS motif containing gene. We have added this information to the discussion (lines 336-345).

• Lines 170-172: While the authors state that cilia look grossly normal, additional quantitative metrics would strengthen their argument. Are cilia lengths normal? Is there some way to assess cilia polarity or the degree of splaying that occurs in other mutants? While the authors are likely correct, having additional evidence that is less subjective would add rigor.

We have added quantification of cilia length for all cell types we examined. This is in the new version of Figure 1.

• To more fully support the argument that alms1 may have a role in auditory hair cells, the authors should examine cilia during auditory hair cell development or maintenance in alms1 larvae by imaging cilia within the otic vesicle (see early work from Jarema Malicki or Bruce Riley for examples).

We imaged the otic vesicle during development as the viewer suggested and included this in the new version of Figure 1.

Minor concerns

• Line 127: please correct the dilution value for the otoferlin antibody.

We have corrected this. The antibody was diluted at 1:100. 

Reviewer #2

- Even if the final message of this publication is the lack of phenotype, one concern is the absence of additional analysis. As an example, in figure 1, I would appreciate to know the number of ciliated cells even if change will not be observed. Also, authors should draw arrows to point the lateral line and cells of the olfactory pit in both control and Alms1 mutant zebrafish.

We have quantified the length of the cilia and noted the number of animals we did this in. The close contact of the cells we are examining at 5dpf make it difficult to distinguish which cilia come from which cell to get an exact cilia/cell count. However, in the 24 hpf fish where we are only looking at 2 ciliated hair cells in the otic vesicle we do not note any cells without cilia which we have noted in the results section (lines 191-193).

We have also added arrows and brackets to figure 1 to label the cilia of the hair cells in 1A & 1B. We were not sure how to do this for the olfactory pit as the entire structure imaged is the olfactory pit.

- Microscopy material and method details associated with the antibody staining are lacking in the method paragraph. Number of cells/cilia counted should also be precise.

We have changed the “antibody labeling” methods section to “antibody labeling, imaging, and cilia length measurement” and included details about how the images were generated (lines 130-133). As we have quantified cilia length as mentioned above we also have the information as to the number of animals we did that for in the figure legend for that data.

- Escaping approach used to study mechanotransduction is not described in the method and no quantification of the escaping movement is provided with the supplemental video.

We have added a section in the methods on the techniques used for “vibrational stimulus response” (lines 237-241). Additionally, we have quantified the number of fish that show a response and included this as a new Figure 2. 

- Discussion paragraph on the lack of conservation with the N terminal mammalian isoform is pretty unclear and might need to be clarified or reformulated.

We have rewritten that section as follows “It is also possible that the lack of phenotype we observe is due to a lack of sequence conservation between the mammalian and zebrafish alms1 gene. The majority of the N-terminus of the human ALMS1 gene, encoding for the tandem repeat domain of the protein, is specific to mammals [67,68] with the first 1560 amino acids of the human gene having no homologous region in the zebrafish gene.” (lines 324-331)

---

## [Decision Letter · Decision Letter 1]

27 Jan 2021

alms1 mutant zebrafish do not show hair cell phenotypes seen in other cilia mutants

PONE-D-20-38618R1

Dear Dr. Stawicki,

We’re pleased to inform you that your manuscript has been judged scientifically suitable for publication and will be formally accepted for publication once it meets all outstanding technical requirements.

Kind regards,

Hemant Khanna

Academic Editor

PLOS ONE

Additional Editor Comments (optional):

Reviewers' comments:

Reviewer's Responses to Questions

**Comments to the Author**

1. If the authors have adequately addressed your comments raised in a previous round of review and you feel that this manuscript is now acceptable for publication, you may indicate that here to bypass the “Comments to the Author” section, enter your conflict of interest statement in the “Confidential to Editor” section, and submit your "Accept" recommendation.

Reviewer #1: All comments have been addressed

2. Is the manuscript technically sound, and do the data support the conclusions?

Reviewer #1: Yes

3. Has the statistical analysis been performed appropriately and rigorously? 

Reviewer #1: Yes

4. Have the authors made all data underlying the findings in their manuscript fully available?

Reviewer #1: Yes

5. Is the manuscript presented in an intelligible fashion and written in standard English?

Reviewer #1: Yes

6. Review Comments to the Author

Reviewer #1: (No Response)

7. PLOS authors have the option to publish the peer review history of their article (what does this mean?). If published, this will include your full peer review and any attached files.

Reviewer #1: No

---

## [Editor Report · Acceptance letter]

22 Mar 2021

PONE-D-20-38618R1 

*alms1* mutant zebrafish do not show hair cell phenotypes seen in other cilia mutants 

Dear Dr. Stawicki:

I'm pleased to inform you that your manuscript has been deemed suitable for publication in PLOS ONE. Congratulations! Your manuscript is now with our production department. 

Kind regards, 

on behalf of

Dr. Hemant Khanna 

Academic Editor

PLOS ONE